# A novel approach to studying infective endocarditis: Ultrasound-guided wire injury and bacterial challenge in mice

**Benedikt Bartsch**[1]*, **Ansgar Ackerschott**[1], **Muntadher Al Zaidi**[1], **Raul Nicolas Jamin**[1], **Mariam Louis Fathy Nazir**[2], **Moritz Altrogge**[2], **Lars Fester**[3], **Jessica Lambertz**[3], **Mark Coburn**[2], **Georg Nickenig**[1], **Marijo Parcina**[4], **Sebastian Zimmer**[1], **Christina Katharina Weisheit**[2]

1 Department of Internal Medicine-II, Heart Center Bonn, University Hospital Bonn, Bonn, Germany,
2 Department of Anesthesiology and Intensive Care Medicine, University Hospital Bonn, Bonn, Germany,
3 Institute of Neuroanatomy of the University of Bonn, University Bonn, Bonn, Germany, 4 Institute of Medical Microbiology, Immunology and Parasitology (IMMIP), University Hospital Bonn, Bonn, Germany

* benedikt.bartsch@ukbonn.de

## Abstract

### Background

Infective endocarditis (IE) is frequently caused by Staphylococcus aureus (S. aureus) and most commonly affects the aortic valve. Early diagnosis and treatment initiation are challenging because the involved immunological processes are poorly understood due to a lack of suitable *in vivo* models.

### Objectives

To establish a novel reproducible murine IE model, based on ultrasound-guided wire injury (WI) induced endothelial damage.

### Methods

IE was established by inducing endothelial damage via ultrasound-guided wire injury followed by bacterial challenge with S. aureus using $10^{4-6}$ colony-forming units (CFU) 24h to 72h after wire injury. Cross-sections of valvular leaflets were prepared for scanning electron microscopy (SEM) and immunofluorescence microscopy to visualize valvular invasion of macrophages, neutrophils, and S. aureus. Bacterial cultivation was carried out from blood and valve samples. Systemic immune response was assessed using flow cytometry.

### Results

Wire injury induced endothelial damage was observed in all mice after wire-injury in SEM imaging. We reliably induced IE using $10^5$ (85%) and $10^6$ (91%) CFU S. aureus after wire injury. Aortic regurgitation was more prevalent in wire injury mice after bacterial challenge. Mice undergoing bacterial challenge responded with significant neutrophilia and elevated pro-inflammatory cytokines in the blood. Immunofluorescence staining revealed significantly increased immune cell accumulations using our proposed model compared to controls.

**Data availability statement:** All relevant data are within the paper and its Supporting Information files.

**Funding:** BB was funded by BONFOR-Gerok-Grant (No.: O-109.0073). CKW was funded by the Deutsche Forschungsgemeinschaft (grant No.: 535107899), GN, SZ and CKW are funded by the Deutsche Forschungsgemeinschaft (DFG, German Research Foundation) – Grant No. 397484323 – Project number 426093965. SZ and CKW are members of the excellence cluster "ImmunoSensation" at Bonn University. We declare that the funders had no role in study design, data collection and analysis, decision to publish, or preparation of the manuscript.

**Competing interests:** The authors have no conflicts of interest to declare.

## Conclusion

Echocardiography and *ex vivo* histological staining demonstrated consistent infective endocarditis induction in our new model, combining a wire injury-induced endothelial damage and S. aureus administration. Further exploration of the initial immune cell response and biomarker expression could potentially identify indicators for early IE diagnosis and novel treatment targets.

## Background

Infective endocarditis (IE) is defined as a bacterial infection of the heart, most commonly affecting the heart valves [1]. Its incidence rose in developed countries from 124,759 cases in 1990 to 251,565 cases in 2019 [2–4]. Despite optimal medical therapy and a trend towards early surgical intervention, mortality rates remain high [5]. Major complications include septic stroke or acute heart failure [2,5]. Patients initially present with only mild symptoms and no direct IE-specific surrogates; echocardiography, CT or PET-CT scans are often inconclusive, hindering early diagnosis and treatment [2]. Risk factors for IE include cardiac procedures such as valve replacement or pre-existing innate or degenerative valvular disease [2,6]. The most common bacteria causing IE in developed countries is Staphylococcus aureus (S. aureus), and its incidence is increasing [1]. The aortic valve is the most commonly affected valve in IE due to the prevalence of aortic valve stenosis (AS) in the elderly with up to 2% [7,8]. While its progression is often slow and asymptotic, severe AS has a mortality rate in the range of 50% if untreated [9].

Our understanding of the mechanisms that cause IE and inflammatory responses is still limited. Bacteriemia is considered a precursor of both native and prosthetic IE [10–12]. Healthy heart valves have a natural barrier to bacterial adhesion and infiltration consisting of an intact endothelial layer, making patients with previous valvular pathology such as stenosis or rheumatic fever susceptible to bacterial metastasis [12].

Endothelial damage exposes the underlying tissue of collagen matrix and interstitial cells to the bloodstream, resulting in fibrin and von Willebrand factor (vWF) adhesion and platelet activation, leading to microthrombotic lesions [13]. During episodes of bacteremia, these lesions act as gateways for bacterial infiltration. S. aureus expresses abundant adhesive surface proteins, making it an ideal pathogen to infiltrate valves with prior endothelial damage [14]. After valve infiltration, S. aureus promotes further platelet activation with its surface proteins [15]. Platelet activation is facilitated by soluble fibrin within damaged valves, which creates a protective shield that impedes immune cell infiltration and serves as an ideal environment for bacterial adhesion [13,16]. While further direct immune cell interaction with the bacteria is impeded by the fibrin-platelet coat, monocytes, macrophages and neutrophils were shown to infiltrate IE valves [17]. IE patients often show an increase of neutrophils in peripheral blood counts which normalizes after treatment initiation. To further evade the immune response, S. aureus can infiltrate both, endothelial cells and fibroblasts using FnBPs [12].

To date, existing murine models of IE either lacked an adequate immune response, making quantification or localization analysis difficult, or exhibited signs of fulminant septic shock, including valvular abscess formation or potential cardiogenic shock due to blunt aortic leaflet injury [13,18,19]. To date, most models rely on either a single i.v. injection (between $10^{5-7}$ CFU) of bacteria or permanent placement of a 32-G catheter across the aortic valve, ignoring the most common clinical setting in which IE develops within pre-existing endothelial damage of the heart valves (e.g., AS). We have recently introduced a novel murine model of AS, which allows us to reliably induce AS in mice [20]. In this study, we aim to establish a murine

model of IE that mimics the most common clinical setting: endothelial damage followed by bacteremia via i.v. S. aureus injection.

## Methods

### Mice

The mice used in this study were male and female C57BL/6-J (wild-type), aged between 10 to 12 weeks, and obtained from Janvier Labs, France. The study was conducted in accordance with the guidelines set by the Animal Ethics Committee of the North Rhine-Westphalian State Agency for Nature, Environment, and Consumer Protection in Germany (AZ 81-02.04.2020.A174). Blood and heart samples were collected under general anesthesia (Xylazin-Hydrochlorid (16mg/kg body weight) and Ketamin-Hydrochlorid (100mg/kg body weight) via intraperitoneal injection. The successful anesthesia was clinically confirmed by the absence of pain reflexes in both the forelimbs and hindlimbs.

### Echocardiography

For the assessment of valvular function, mice were sedated with 1.5% isoflurane under strict surveillance of respiratory rate, electrocardiogram, and body temperature. Aortic valve peak velocity, aortic regurgitation, and bacterial valve vegetations were measured in the suprasternal view (S1 Fig). Other standard echocardiographic parameters such as left ventricular ejection fraction, fractional shortening, and ventricular volumes were measured in the parasternal long-axis views. Regurgitation jets of the aortic valve were defined as follows: in immediate proximity to the aortic valve: mild; if jets reached the length of the left ventricular outflow tract: moderate; and if jets reached beyond the outflow tract: severe. An analysis of the echocardiography data was performed by the investigator after blinding and randomization. Most common findings of IE in echocardiography in the clinical setting are increases in aortic valve cusp diameter, presence of aortic regurgitation and an increase in ventricular volumes. Aortic peak velocity assesses aortic valve function is increased when AS is present.

### Wire injury

The wire injury procedure was conducted following a recently published protocol from our facility (Niepmann et al., 2019). Animals were kept at our facility at least one week prior to experiment start to reduce animal stress levels. Mice were anesthetized using Xylazin-Hydrochlorid (16mg/kg body weight) and Ketamin-Hydrochlorid (100mg/kg body weight) via intraperitoneal injection. In brief, the right carotid artery was surgically accessed, and a straight guidewire with a shortened and soldered tip (Abbott HI-TORQUE 0.014) was used. Blood flow was halted with the application of two ligatures. To induce endothelial damage, the wire was moved back and forth across the valve 50 times and rotated 100 times. During the procedure, echocardiography was immediately performed to ensure that no aortic regurgitation occurred. Subsequently, the wire was removed, and the artery was ligated.

### Bacterial challenge and cultivation

The methicillin-susceptible S. aureus strain SA-LT 68/03C12Y7, isolated from human IE samples, was preserved at -80°C in 20% glycerol. To create the bacterial challenge suspension, S. aureus was cultured with LB media for 8 hours at 37°C, using a slow, pivoting motion. Subsequently, the bacteria were suspended in sterile 0.9% NaCl to reach a concentration of $10^{4-6}$ CFU/100 μl. Prior to the experiment, the appropriate bacterial volumes were confirmed by inoculating 10 μl of bacteria on 5% blood agar plates (BD Biosciences). Blood and

homogenized valve samples were inoculated on 5% blood agar plates as well. The bacterial challenge was carried out either 1 day or 3 days after the wire injury by intravenously injecting 0.1 ml of the bacterial suspension.

### RT-PCR

RT-PCR from aortic valve cross sections was performed using primers for Enterotoxin (SEC-1 5′-GACATAAAAGCTAGGAATTT3′; SEC-2 5′-AATCGGATTAACATTATCC-3′) and Alpha-Toxin (Hla-PCR-F1,5′-TGTCTCAACTGCATTATTCTAAATTG-3′, Hla-PCR-R1,5′-CATCATTTCTGATGTTATCGGCTA-3′ in our strain [21].

### Plasma samples

Serum samples were obtained immediately before sacrifice. Sera were stored at −80°C until the time of analysis. IL-1α, IL-1β, IL-10 and M-CSF were measure using Mouse XL Cytokine Luminex® Performance Premixed Kit. The manufacturer's (R&D Systems) recommended protocol was followed.

### Histological preparation

Mice were euthanized via cervical dislocation 1d, 3d, or 7d after the bacterial challenge. Hearts were extracted using sterile instruments and were flushed with sterile 0.9% saline solution. Hearts where then either purified for bacterial culture analysis using Precellys® or made available to immunofluorescence.

CD68 and CD45 staining was used to visualize immune cell and macrophage infiltration of the aortic valve as described in Niepmann et al 2019. For Ly6G and S. aureus staining, aortic valve sections were fixed in Acetone for 30 min and blocked with 10% normal goat serum (NGS). The primary antibody was diluted 1:200 (anti Ly6G, rat 1A8 anti mouse, BD Biosciences, USA, Anti-S. aureus IgG rabbit, ab20920, Abcam, UK) and incubated overnight. The secondary antibody was diluted 1:500 (Cy3 AffiniPure Donkey anti Rat IgG, Jackson ImmunoResearch Laboratories Inc, AlexaFluor 647, goat anti rabbit, ThermoFisher Scientific, USA) and incubated for 90 minutes.

### Flow cytometry

For quantification of immune responses, 50 μl of whole blood was collected. Cells were stained using anti-Ly6G (BioLegend), anti-B220 (BioLegend), anti-CD11b (BioLegend), anti-CD3 (eBioscience), CD45 (BioLegend) and anti NK1.1 (BioLegend). For flow cytometry we used FACSCanto II (BD Bioscience, Franklin Lakes, USA) and analyzed the data with FlowJo (Tree Star, Ashland, USA).

### Scanning electron microscopy (SEM)

SEM imaging was performed in a laboratory outside of the animal facility. The imagers were blinded to the animal treatment and were only given numbered valvular samples that did not contain any information about animal treatment. Samples underwent fixation through transcardiac perfusion with 6% glutaraldehyde in phosphate buffer. Dehydration was achieved using a graded ethanol series, followed by a transition to acetone. Afterward, this intermediate step was eliminated using a critical point dryer, and a coating of 8 nm of platinum was applied using a sputter coater, prior to visualization using a scanning electron microscope (Jeol 7500F). The examiners were blinded to the intervention and analyzed multiple regions per sample (6 regions per valve).

## Statistical analysis

The data presented is shown as mean ± SEM. Appropriate assumptions of data (e.g., normal distribution or similar variation between experimental groups) were examined before statistical tests were conducted. The number of experiments and the number of mice per group are provided in the figure legends. Student's t-tests were used whenever two groups were compared, and one-way or two-way analyses of variance followed by Tukey's test for multiple comparisons. The analysis was performed with Prism 8 (GraphPad Software, Inc. La Jolla, CA). The results are provided as mean unless noted otherwise; $p < 0.05$ was considered statistically significant. Raw data used for statistical analysis can be found in the data file in Supporting Information.

## Results

### Wire injury + bacterial challenge mice showed increased bacteremia and valvular infiltration levels of S. aureus

Since bacteremia is considered a necessary precursor in IE development, we first tested bacterial challenge via intravenous injection in wild-type C57BL/6-J mice using $10^4$, $10^5$ and $10^6$ CFU S. aureus for spontaneous development of IE (Fig 1a-c). The overall success rate of IE induction confirmed with valvular cultures was 22% ($10^4$ CFU), 27% ($10^5$ CFU) and 67% ($10^6$ CFU) (Fig 1d). Mice receiving $10^6$ CFU bacterial challenge suffered severe morbidity and clinical signs of sepsis including weight loss, reduced mobility and decreased fur grooming.

The immunological response following bacterial challenge using concentrations of $10^5$ and $10^6$ CFU was verified by increased neutrophil counts (S1 Table).

Because a singular bacterial challenge at concentrations below the sepsis threshold was unreliable in inducing endocarditis, we extended our experimental protocol to include a wire injury of the aortic valve before bacterial challenge to create endothelial damage (Fig 1e). Bacterial challenge was performed either 24h or 72h after wire injury. Blood cultures and cultures from valvular samples revealed dose dependent blood bacterial titers and valvular bacterial colonies (Fig 1g + h). While blood cultures and IE induction success rate were significantly increased using $10^5$ and $10^6$ CFU compared to $10^4$ CFU, there was no difference in blood cultures when comparing $10^5$ and $10^6$ CFU among wire injury (WI + BC) animals and IE induction rate plateaued after using $10^5$ CFU. IE induction rate was 22% ($10^4$ CFU), 86% ($10^5$ CFU) and 92% ($10^6$ CFU) (Fig 1f).

The highest absolute and relative number of neutrophils was detected in the WI + BC group 1d after wire injury + bacterial challenge (213613 ± 40032 cells, 71,6% ± 8.7%, n = 18) (see supplements).

Overall mortality for WI + BC mice was 13.8%, 80% of deaths occurred in WI + BC with $10^6$ CFU. Since sepsis frequency and mortality were elevated in $10^6$ CFU, while $10^5$ CFU could reliably induce bacteremia and valvular infiltration, we proceeded using $10^5$ CFU for our further investigations.

Valvular S. aureus infiltration was increased in WI + BC mice at day 3 and 7 after bacterial challenge compared to BC only mice (Fig 1i). We found a difference in S. aureus infiltration depending on the timing of bacterial challenge after wire injury (24h vs 72h) in WI + BC mice, favoring bacterial challenge after 72h (see supplements).

### Wire injury leads to AS development and causes endothelial damage

We observed endothelial damage and superficial fibrin depositions and bacterial infiltration in all WI + BC samples (6/6), while the endothelium was intact in all BC samples (6/6) (Fig 2a-c). Bacterial infiltration (violet) could be detected in areas of endothelial damage only (Fig 2e-f).

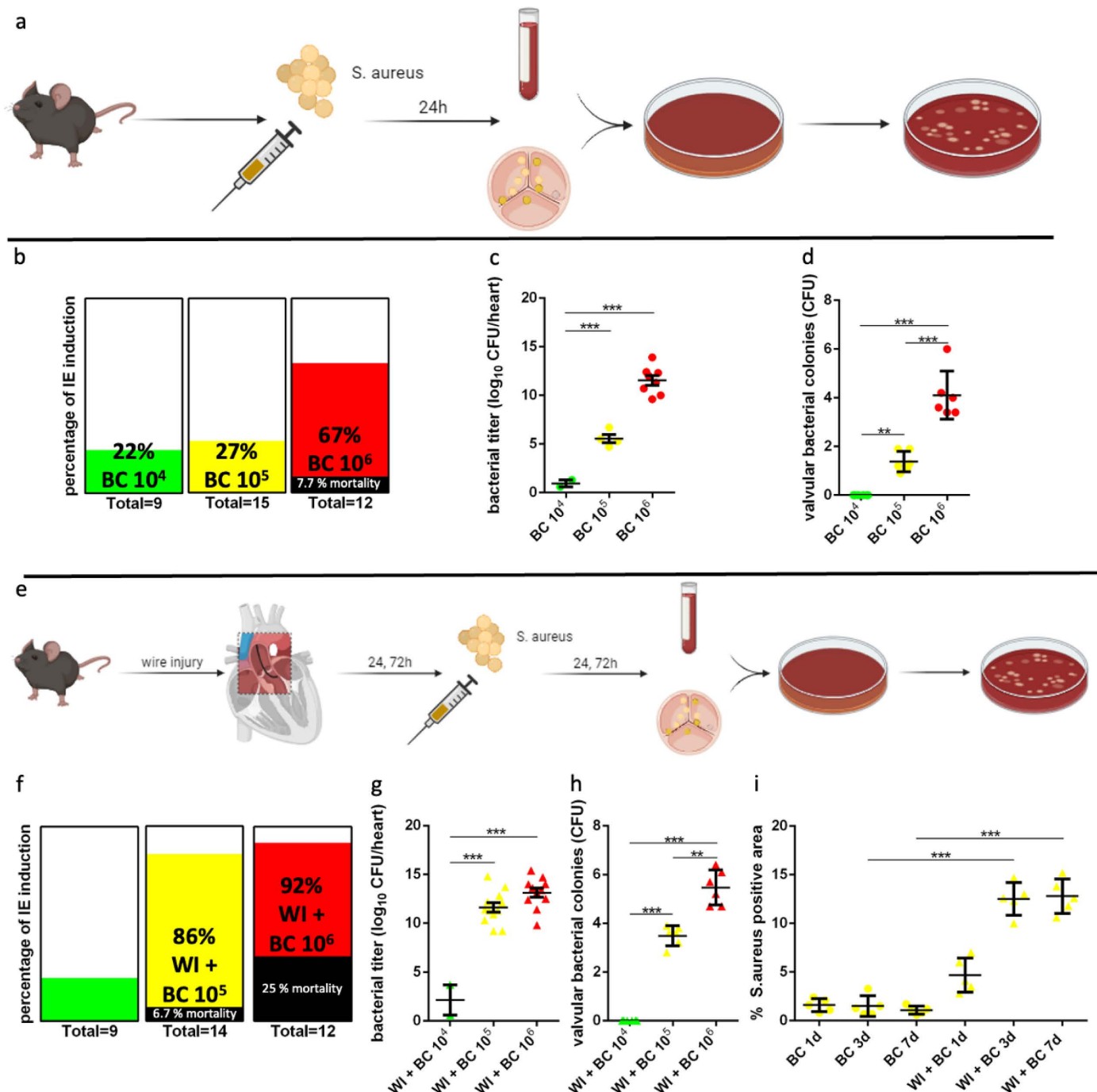

**Fig 1. Experimental protocol and bacterial growth in mice after bacterial challenge.** (a) Bacterial challenge was performed using $10^4$, $10^5$, and $10^6$ CFU for i.v. injection. Blood and valve samples were collected 24h after bacterial challenge and microbiologically examined. (b-d) Successful IE induction was determined 24h after bacterial challenge in %. CFU concentration in blood and valvular samples was measured 18h after incubation using agar. (e) Mice underwent wire injury (WI) to induce endothelial damage, bacterial challenge was performed either 24h or 72h after WI using $10^4$, $10^5$, and $10^6$ CFU, blood and valve samples were collected 24h and 72h after bacterial challenge. (f-h) Successful IE induction was determined 24h after bacterial challenge in %. CFU concentration in blood and valvular samples was measured 18h after incubation using agar. (i) Immunofluorescence microscopy data for S. aureus. Data in the quantitative plots are presented as mean ± SEM, and statistical significance was determined using unpaired one-way ANOVA. ***$P < 0.001$; **$P < 0.01$; *$P < 0.05$. BC = bacterial challenge.

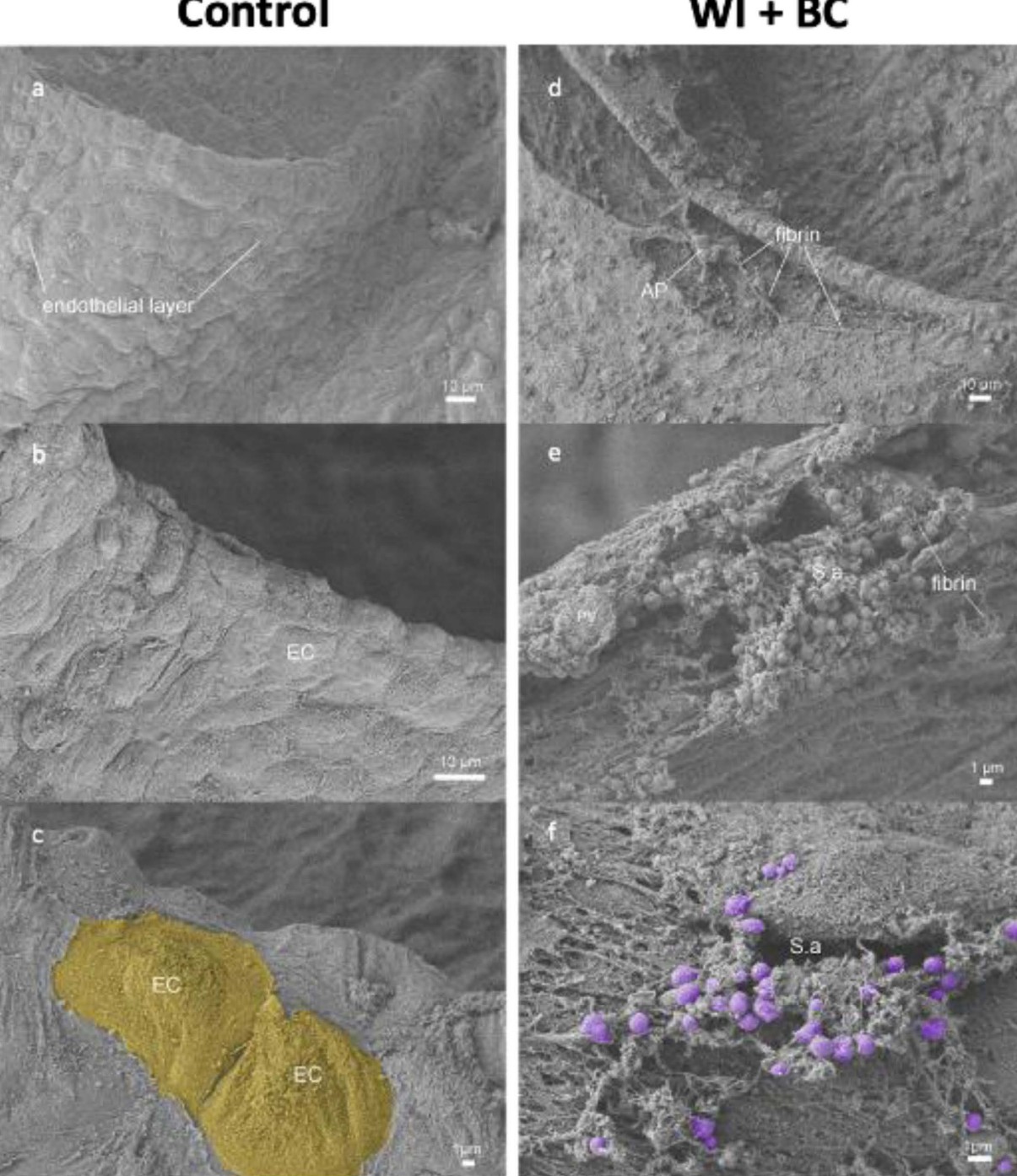

**Fig 2. Scanning electron microscopy after bacterial challenge.** (a-c) Scanning electron microscopy images (2.00 KV LEI, 14mm WD) of aortic valve leaflet cross-sections of BC animals three days after bacterial challenge. (d-f) Wire injury causes endothelial damage with only few endothelial cells remaining (EC) and enables S. aureus (S.a., violet) infiltration via fibrin layers and activated platelets (AP). BC = bacterial challenge, WI = wire injury.

### Wire injury + bacterial challenge mice were prone to bacterial vegetation in transthoracic echocardiography

6d after wire injury (3d after bacterial challenge), transthoracic echocardiography was conducted. Aortic peak velocity (Fig 3d) was increased in WI + BC compared to BC mice. Aortic regurgitation was present among BC animals in one mouse using $10^6$ CFU (12.5%, n = 8). Among WI + BC mice we detected mild regurgitation in 75%, and moderate regurgitation in 12.5%, with no aortic regurgitation present in 12.5% of mice using $10^5$ S. aureus (n = 8) (Fig 3a). After BC with $10^6$ CFU aortic regurgitation was detectable in all WI + BC mice, 12.5% showed mild, 75% moderate and 12.5% severe signs of aortic regurgitation. End-diastolic and end-systolic volumes were increased in WI + BC mice compared to BC mice (Fig 3d).

To assess aortic valve vegetation, we measured the AV cusp thickness in both parasternal long and short axis views. WI + BC mice showed thicker AV cusps compared to BC only mice (Fig 3b-d).

### Bacterial challenge after wire injury promotes valvular immune cell infiltration and sustained inflammation in plasma samples

We performed immunofluorescence staining using CD45, CD68, and Ly6G (Fig 4) to assess immune cell infiltration at day 1, 3, or 7 after bacterial challenge with $10^5$ CFU. CD45$^+$, CD68$^+$ and Ly6G$^+$ infiltration was higher in WI + BC mice after 3 and 7 days compared to BC mice (Fig 4a). However, CD68$^+$ macrophage infiltration was already elevated in WI + BC compared to BC mice 1d after bacterial challenge (Fig 4b).

To further assess systemic inflammation we analyzed pro-inflammatory cytokines IL-1α, IL-1β, IL-10 and due to the increased valvular infiltration of macrophages we measured Macrophage colony-stimulating factor (M-CSF) 3d after bacterial challenge using $10^5$ CFU.

While increased expression patterns of pro-inflammatory cytokines IL-1α, IL-1β, IL-10 (Fig 4d-g) were observed in the BC group and the WI + BC group three days after bacterial administration, these effects were significantly more pronounced in the WI + BC group. M-CSF was increased in WI + BC animals, while we found no difference between BC only and control animals.

### Effects of S. aureus toxins on valvular infiltration

To analyze the effect of S. aureus toxins on valvular infiltration we analyzed the most dominant toxins expressed in our strain (SA-LT 68/03C12Y7) Enterotoxin and alpha-Toxin in valvular sections of murine samples with positive S. aureus infiltration in immunofluorescence analysis (Fig 1i). The analysis of enterotoxin revealed no relevant differences between the BC and WI+BC groups, nor were any changes observed over time (1 day vs. 3 days post-BC) (Fig 5a). Similarly, the analysis of alpha-toxin showed no differences between the BC and WI+BC groups. However, in the BC group, higher endotoxin levels were detected at 3d compared to 1d post-challenge (Fig 5b).

## Discussion

In this study, we successfully established a new aortic valve IE model combining a wire injury-induced endothelial damage model and i.v. bacterial challenge with S. aureus. To confirm IE induction, we employed two independent methods: valvular cultures and S. aureus immunofluorescence. The latter method, in addition to pure quantification, also allows localization of the bacteria on the valve and identification of abscesses; therefore, an observation period of up to 7 days was applied. However, we found no differences in bacterial localization over time, nor did we observe the occurrence of aortitis in the proximal aorta.

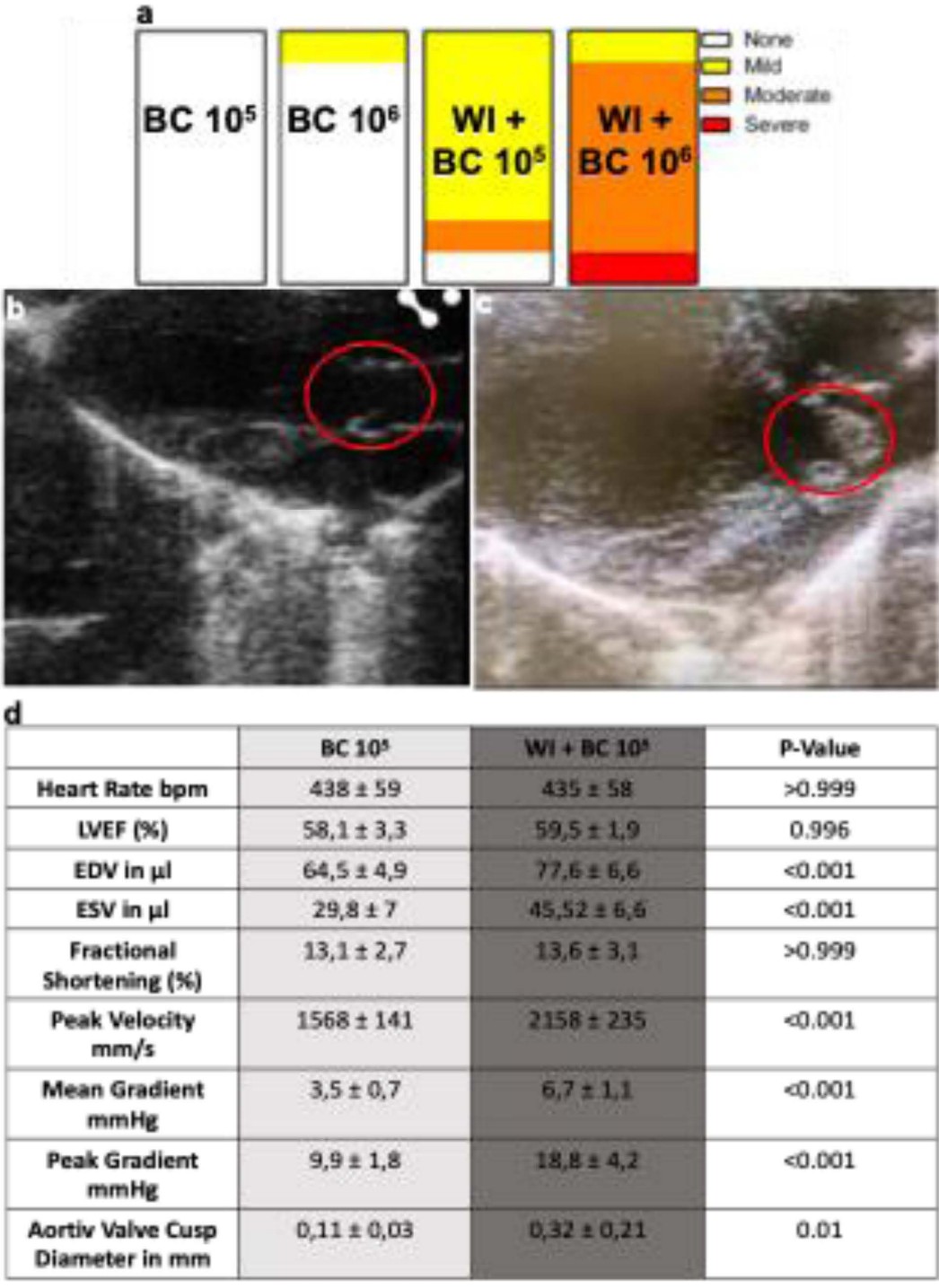

**Fig 3. Echocardiographic analysis.** (a) Aortic regurgitation was more prevalent in mice after wire injury and bacterial challenge. (b-c) Exemplary images of a healty aortic valve (b) and aortic valve endocarditis (c) vegetations in parasternal long-axis view after wire injury and bacterial challenge. (d) Left ventricular ejection fraction (LVEF), left ventricular volumes and fractional shortening (%) were measured in parasternal long-axis view after wire injury and bacterial challenge. Data is presented as mean ± SEM, and statistical significance was determined using unpaired one-way ANOVA. ***P < 0.001; **P < 0.01; *P < 0.05. BC = bacterial challenge, WI = wire injury.

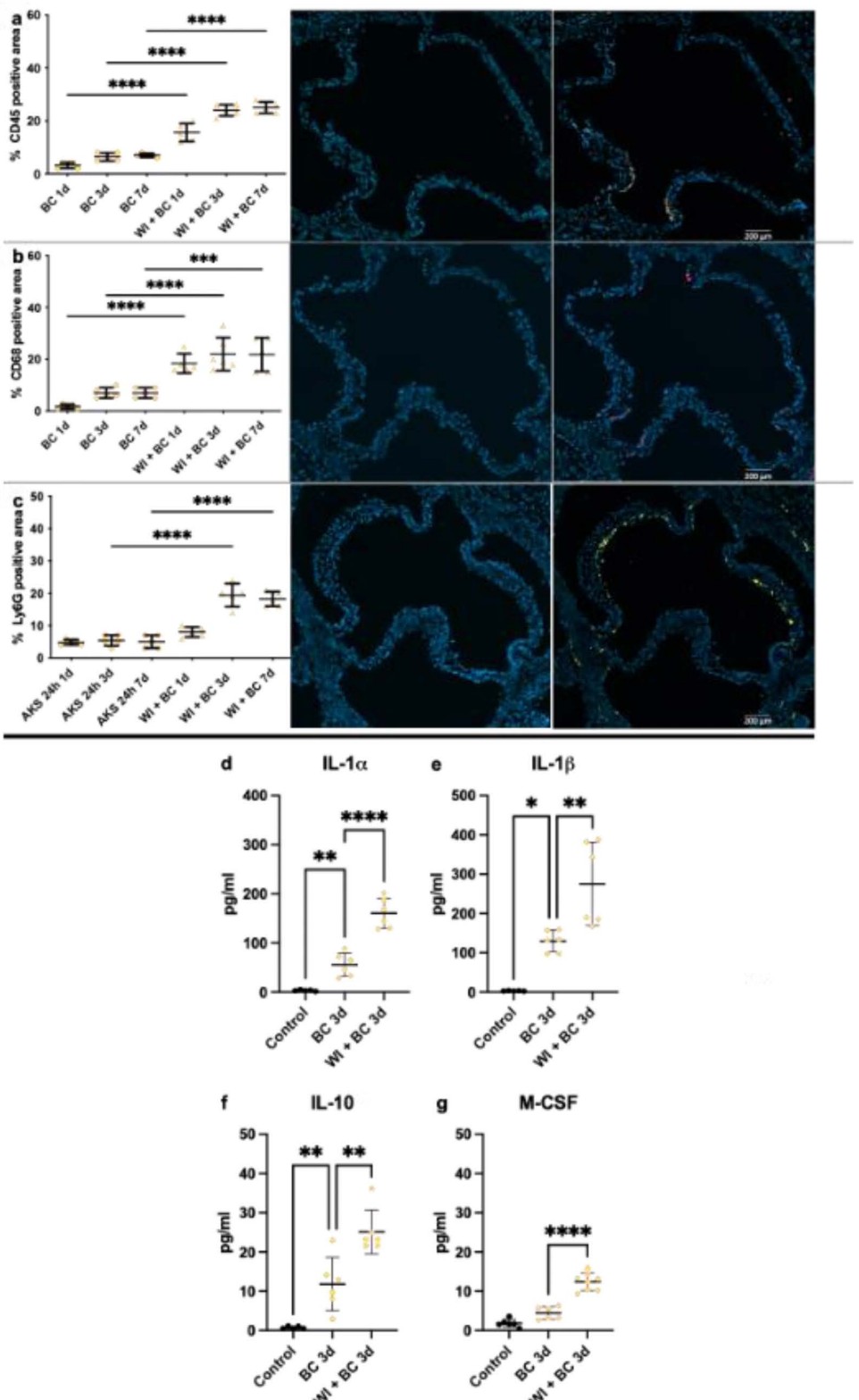

**Fig 4. Immunofluorescence staining and cytokine expression levels.** (a-c) Immunofluorescence microscopy was performed after mice were sacrificed and the hearts were collected. Representative images and quantitative analysis of CD45 (a), CD68 (b), Ly6G (c) in WI + BC mice. (d-g) Pro-inflammatory cytokines IL-1α, IL-1β, IL-10 and

macrophage-colony stimulating-factor (M-CSF) were measured from sera taken immediately before sacrifice 3d after bacterial. Data in the quantitative plots are presented as mean ± SEM, and statistical significance was determined using unpaired one-way ANOVA. ***P < 0.001; **P < 0.01; *P < 0.05. BC = bacterial challenge, WI = wire injury.

Wire injury showed distinctive features of endothelial damage in SEM, including fibrin depositions, that could serve as an ideal entry for S. aureus. Bacteremia and valvular infiltration of S. aureus was dose dependent with $10^6$ CFU showing the highest titers. The S. aureus ($10^{5-6}$ CFU) concentrations used to successfully induce IE were similar or lower to groups with different proposed IE models. $10^4$ CFU of S. aureus were unable to induce IE in both groups and IE induction plateaued between $10^5$ and $10^6$ CFU of S. aureus. Since all mice in the SEM showed evidence of endothelial damage and were thus susceptible to IE development, it can be assumed that immunomodulatory effects influenced the animals that did not develop endocarditis. These effects should be further explored in the future, including the impact of the microbiome, autoreactive effects, and early immune response activation [22–24]. Using the concentrations proposed in other studies we found higher mortality rates due to peripheral bacterial spreading including peripheral abscess formation and sepsis. These observations make it difficult to interpret immune cell responses as IE specific, especially when trying to potentially use early inflammation patterns as a prerequisite for IE detection using imaging techniques [13,18,19]. Accounting for its high mortality rate $10^6$ CFU as BC with high neutrophilia and yet similar bacterial vegetations proved to be more suitable for detecting severe endocarditis including sepsis compared to $10^5$ CFU.

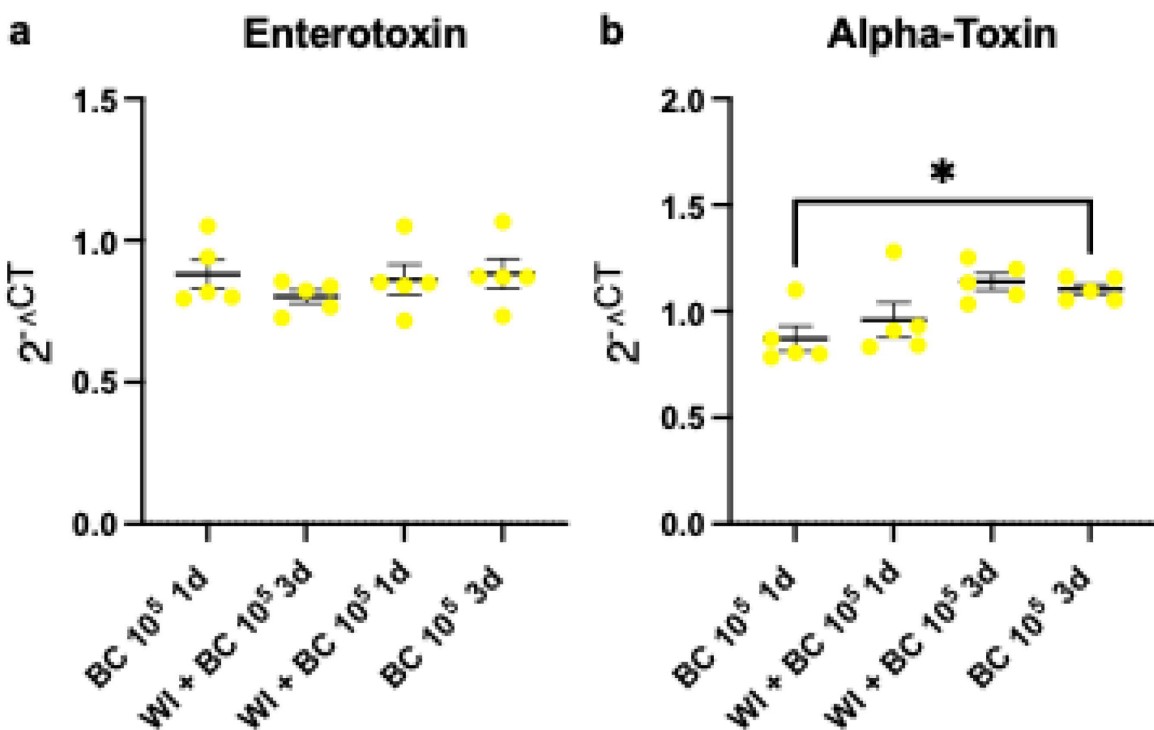

**Fig 5. Enterotoxin and Alpha-Toxin levels 1d and 3d after bacterial challenge.** (a) There was no difference in Enterotoxin expression between BC and WI+BC mice and no difference between different time points. (b) Alpha-Toxin was not differently expressed between BC and WI+BC, however Alpha-Toxin was overexpressed in BC animals after 3d compared to 1d. Data in the quantitative plots are presented as mean ± SEM, and statistical significance was determined using unpaired one-way ANOVA. ***P < 0.001; **P < 0.01; *P < 0.05. BC = bacterial challenge, WI = wire injury.

As previously published data from our group indicate, we could detect AS one week after wire injury via echocardiography (increase in aortic peak velocity). After bacterial challenge we found an increase in aortic valve cusp diameter *in-vivo* as sign of bacterial vegetations, and an increase in ventricular volumes (EDV and ESV) and aortic regurgitation (Fig 3a) frequency as well as severity, especially when using $10^6$ CFU similar to findings in patients suffering from IE [25]. All immune cell markers (CD45, CD68 and Ly6G) and pro-inflammatory cytokines in blood samples were increased in WI + BC mice compared to BC mice. Other studies, using different IE models, have not yet investigated immune cell response in blood or valvular tissue [13,18,19]. Studies in human samples found similar cellular infiltration pattern as we did in our murine model, stressing the close link between our model and human pathophysiology [17].

Most previously published models rely on inducing valvular damage by placing a catheter across the aortic valve and performing bacterial challenge afterwards with concentrations ranging between $10^{5-6}$ CFU [13,18,26,27]. In these models the catheter scratches the endothelium with every heartbeat in an uncontrolled manner potentially causing acute regurgitation and cardiogenic shock. Endothelial damage was not evaluated in these studies. Using SEM to detect endothelial viability has been tested on human samples but was not used in an IE setting [28,29]. Due to its high-resolution SEM allowed for specific assessment of endothelial damage and simultaneous assessment of direct bacterial infiltration and its interaction with cellular proteins such as fibrin (Fig 2). Valvular function after operation was never assessed in any of the previous models. Our group detected during the establishment of our wire-injury model that uncontrolled blunt trauma without ultrasound guidance to the aortic valve leaflets not only induces endothelial damage but often causes severe aortic regurgitation and cardiogenic shock making it unsuitable to further assess IE pathophysiology and immune response as it may overlap with cardiogenic shock [20,30].

In other models of IE, the observation period after bacterial challenge was limited to 3 days and bacteremia was performed simultaneously with wire placement [13,18,19]. With our study we constructed a murine IE model that resembles the most common two-phasic pathophysiology of human IE consisting of an initial valvular damage followed by bacteremia after several hours and days without inducing cardiogenic shock due to blunt valve trauma. The striking differences between successful IE induction between WI + BC mice and BC mice highlights the significance of endothelial damage as a key requisite for IE induction and further underlines the proximity of our murine model in comparison with the human pathophysiology [12,31,32]. Interestingly, we did not find an overexpression of Enterotoxin and alpha-Toxin in valvular samples with successful IE induction. The immunosuppressive effects of bacterial endotoxins have been well-documented, particularly in vitro [33]. While our strain did not demonstrate overexpression of enterotoxins or alpha-toxin, it cannot be ruled out that these factors might play a role during the acute phase immediately following the bacterial challenge.

Due to its experimental design certain limitations apply. Murine and human biology and its response to bacterial inflammation differ between species. Acute endothelial damage induction via wire injury differs from human AS development as a chronic condition, taking years to develop. While the S. aureus strains used in this experiment was isolated from human IE samples the efficacy of this model need to be evaluated in other strains as well.

The wire injury-based IE model established in this study provides a reliable, pathophysiological closely linked model of IE in wild-type mice to human IE. Different bacterial concentrations can be used to induce distinct severity grades of IE. To induce IE no genetic alterations or permanent wire must be placed, thus it can easily be applied to further analysis of immune cell responses and screen for biomarkers associated with early IE.

## Supporting information

**S1 Table. Flow cytometry from murine blood samples.**
(DOCX)

**S1 Fig. Exemplary images of echogradiographic analysis.**
(DOCX)

**S1 Data. Data file includes all raw data used for analysis.**
(XLSX)

## Author contributions

**Conceptualization:** Benedikt Bartsch, Mariam Louis Fathy Nazir, Sebastian Zimmer, Christina Katharina Weisheit.

**Data curation:** Benedikt Bartsch, Muntadher Al Zaidi, Moritz Altrogge, Jessica Lambertz.

**Formal analysis:** Benedikt Bartsch, Muntadher Al Zaidi, Christina Katharina Weisheit.

**Funding acquisition:** Raul Nicolas Jamin, Mark Coburn, Christina Katharina Weisheit.

**Investigation:** Benedikt Bartsch, Mariam Louis Fathy Nazir, Moritz Altrogge, Jessica Lambertz, Christina Katharina Weisheit.

**Methodology:** Benedikt Bartsch, Marijo Parcina, Sebastian Zimmer, Christina Katharina Weisheit.

**Project administration:** Ansgar Ackerschott, Lars Fester, Mark Coburn.

**Resources:** Benedikt Bartsch, Raul Nicolas Jamin, Georg Nickenig.

**Software:** Benedikt Bartsch, Ansgar Ackerschott, Raul Nicolas Jamin, Sebastian Zimmer.

**Supervision:** Lars Fester, Georg Nickenig, Sebastian Zimmer.

**Validation:** Benedikt Bartsch, Marijo Parcina, Sebastian Zimmer.

**Visualization:** Benedikt Bartsch, Ansgar Ackerschott, Muntadher Al Zaidi, Christina Katharina Weisheit.

**Writing – original draft:** Benedikt Bartsch.

**Writing – review & editing:** Benedikt Bartsch, Ansgar Ackerschott, Sebastian Zimmer, Christina Katharina Weisheit.

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
