## [Decision Letter · Decision Letter 0]

25 Oct 2024

PONE-D-24-31793A novel approach to studying Infective Endocarditis: Ultrasound-guided wire injury and bacterial challenge in micePLOS ONE

Dear Dr. Bartsch,

Thank you for submitting your manuscript to PLOS ONE. After careful consideration, we feel that it has merit but does not fully meet PLOS ONE’s publication criteria as it currently stands. Therefore, we invite you to submit a revised version of the manuscript that addresses the points raised during the review process.

We look forward to receiving your revised manuscript.

Kind regards,

Utpal Sen, Ph.D.

Academic Editor

PLOS ONE

Journal Requirements:

'BB was funded by BONFOR-Gerok-Grant (No.: O-109.0073). CKW was funded by the Deutsche Forschungsgemeinschaft (grant No.: 535107899), GN, SZ and CKW are funded by the Deutsche Forschungsgemeinschaft (DFG, German Research Foundation) – Grant No. 397484323 – Project number 426093965. SZ and CKW are members of the excellence cluster ‘‘ImmunoSensation’’ at Bonn University. '

Please state what role the funders took in the study.  If the funders had no role, please state: ''The funders had no role in study design, data collection and analysis, decision to publish, or preparation of the manuscript.'' 

'The authors have no conflicts of interest to declare.'

Please complete your Competing Interests on the online submission form to state any Competing Interests. If you have no competing interests, please state ''The authors have declared that no competing interests exist.'', as detailed online in our guide for authors at http://journals.plos.org/plosone/s/submit-now 

5. Please upload a copy of Figure 5, to which you refer in your text on page 14. If the figure is no longer to be included as part of the submission please remove all reference to it within the text.

6. Please include a copy of Table 1 which you refer to in your text on page 10. 

Reviewers' comments:

Reviewer's Responses to Questions

**Comments to the Author**

1. Is the manuscript technically sound, and do the data support the conclusions?

Reviewer #1: Partly

Reviewer #2: Yes

2. Has the statistical analysis been performed appropriately and rigorously? 

Reviewer #1: Yes

Reviewer #2: Yes

3. Have the authors made all data underlying the findings in their manuscript fully available?

Reviewer #1: Yes

Reviewer #2: Yes

4. Is the manuscript presented in an intelligible fashion and written in standard English?

Reviewer #1: Yes

Reviewer #2: Yes

5. Review Comments to the Author

Reviewer #1: The article presented by Bartsch and colleagues is a well-illustrated and carefully written research study that may be of interest to readers. The authors' efforts to make the article engaging for readers are commendable and warrant recognition.

The three fundamental objectives of this study were successfully accomplished. The initial objective was to develop bacteremia. The second objective was to create a wire lesion to establish a validity criterion for the attachment of Staphylococcus aureus and the subsequent presence of vegetations. The third objective was to demonstrate the presence of vegetations that are detectable on echocardiography.

Nevertheless, an examination of the extant literature on Staphylococcus aureus infection reveals that two points in the manuscript, when considered in the context of the broader research landscape, appear to offer relatively limited value. These points are worthy of further examination.

1. Line 54. The sentence "Infective endocarditis (IE) is defined as a bacterial infection of the heart, most commonly affecting the heart valves (1). Its incidence is increasing in developed countries, and it has high mortality rates of up to 30% (2, 3)" is only partially accurate.

Recently, there has been a decline in the incidence of S. aureus infection for native valve endocarditis, which has been accompanied by an increase in Enterococcus infections (Enterococcus faecalis and Streptococcus gallolyticus) in the same age group.

2. Line 176: Sentence “The overall success rate of confirmed IE induction with valve cultures was 22% (104 CFU), 27% (105 CFU) and 67% 178 (106 CFU) (Fig. 1d). Mice that received a 106 CFU bacterial challenge suffered from severe morbidity and clinical signs of sepsis, including weight loss, reduced mobility, and decreased fur grooming.

It is imperative that the authors elucidate this point, as it has the potential to engender confusion. The severity of a Staphylococcus aureus infection is only partially attributable to the concentration of bacteria involved. It is undoubtedly favored by the production of toxins which have a considerable interference with the immune response. Staphylococcal toxins have been demonstrated to interfere with the cells of both the innate and adaptive immune responses. In particular, toxins such as TSS-1, Staphylococcal endotoxin, and alpha toxin have been shown to be capable of lysing immune cells, including PMN, monocytes, and macrophages that are involved in the clearance of S. aureus. Additionally, these toxins have been observed to impair the function of adaptive immune cells, represented by T and B lymphocytes. Furthermore, the interaction between innate and adaptive immune cells can also be impaired by these toxins.

The authors' assertion would be substantiated if they could provide a detailed account of the degree of cytotoxicity, immune response, and progression in the diameter of the globules for each level of bacterial concentration.

Reviewer #2: The authors present a relatively brief report that describes a new mouse model of aortic valve S. aureus IE model that will in the future be valuable for studying the pathogenesis of disease and/or therapeutic interventions. I have some comments to be addressed in a revised version

1. The Discussion needs to be extended to provide clearer interpretations of the findings in the study

2. The conclusion on line 288 that wire injury mediated endothelial damage is not supported by robust quantitative data. The SEM in Fig 2 shows single images of endothelial damage or fibrin deposition and reports that similar areas were seen in 6 other mice. The authors should clarify how an unbiased and reproducible assessment of endothelial damage or fibrin deposition was carried out in n=6 mice. Do other findings in the study support the conclusion that endothelial damage has occurred

The motivation to use SEM for detecting endothelial damage rather than HnE staining or immunohistochemistry should be included in the text. The strengths and weaknesses of this strategy should be described in the Discussion. Do other studies show similar SEM images of endothelial damage?

2. On lines 105 the authors state that "induction rate was 22% (104 CFU), 86% (105 CFU) and 92% (106 CFU)". The significant hop from 22% to 86% followed by a relative plateau should be discussed and interpreted in the Discussion

3. It is not clear why the immunohistochemistry of S .aureus in Fig 1i is performed up to 7 days which is in contrast to all other time frames in the same Figure. This should be clarified and motivated in text. How can the data in Fig 1g, h and i be interpreted in relation to one another?

4. the wealth of data shown for echocardiography in Fig 3 should be better explained for those not familiar with the technique. Is all data important to include here or can critical parameters be highlighted and focussed on in interpretation of the overall findings?

6. PLOS authors have the option to publish the peer review history of their article (what does this mean? ). If published, this will include your full peer review and any attached files.

**Do you want your identity to be public for this peer review?** For information about this choice, including consent withdrawal, please see our Privacy Policy .

Reviewer #1: No

Reviewer #2: No

---

## [Author Response · Author response to Decision Letter 1]

31 Dec 2024

We thank the reviewers and the editor for their valuable work and the time invested in reviewing our manuscript. We particularly appreciate the suggestions, which have significantly improved the quality of the manuscript.

The stylistic revisions suggested by the editor were implemented in line with the available online guidelines and have been marked accordingly in the text.

Response to Reviewers

Reviewer #1

1. Line 54. The sentence "Infective endocarditis (IE) is defined as a bacterial infection of the heart, most commonly affecting the heart valves (1). Its incidence is increasing in developed countries, and it has high mortality rates of up to 30% (2, 3)" is only partially accurate.

Recently, there has been a decline in the incidence of S. aureus infection for native valve endocarditis, which has been accompanied by an increase in Enterococcus infections (Enterococcus faecalis and Streptococcus gallolyticus) in the same age group.

Response:

We thank the reviewer for the specification. We further defined the epidemiological aspects of IE in developed countries, citing the data from Momtazmanesh et al. 2022 which compared IE epidemiological feature globally between 1990 and 2019. The authors found an increase of IE in high socio-demographic countries from 124,759 cases in 1990 to 251,565 cases in 2019. We specified these demographics in Line 54. The increase is mostly due to an increase in elderly patients and high risk IE groups.

2. Line 176: Sentence “The overall success rate of confirmed IE induction with valve cultures was 22% (104 CFU), 27% (105 CFU) and 67% 178 (106 CFU) (Fig. 1d). Mice that received a 106 CFU bacterial challenge suffered from severe morbidity and clinical signs of sepsis, including weight loss, reduced mobility, and decreased fur grooming.

It is imperative that the authors elucidate this point, as it has the potential to engender confusion. The severity of a Staphylococcus aureus infection is only partially attributable to the concentration of bacteria involved. It is undoubtedly favored by the production of toxins which have a considerable interference with the immune response. Staphylococcal toxins have been demonstrated to interfere with the cells of both the innate and adaptive immune responses. In particular, toxins such as TSS-1, Staphylococcal endotoxin, and alpha toxin have been shown to be capable of lysing immune cells, including PMN, monocytes, and macrophages that are involved in the clearance of S. aureus. Additionally, these toxins have been observed to impair the function of adaptive immune cells, represented by T and B lymphocytes. Furthermore, the interaction between innate and adaptive immune cells can also be impaired by these toxins.

The authors' assertion would be substantiated if they could provide a detailed account of the degree of cytotoxicity, immune response, and progression in the diameter of the globules for each level of bacterial concentration.

Response:

We thank the reviewer for the thoughtful suggestion. We have conducted additional analyses to quantify cytotoxicity. Specifically, we analyzed the expression of toxins produced by the strain utilized in our model (Enterotoxin + Alpha-Toxin) for the respective time-points (Fig. 5a-b) measuring aortic valve cross section samples of our animals. The results show a significant increase in alpha-toxin comparing day 1 and day 3. There is no increase in Enterotoxin levels detectable over time These data have been included in the revised manuscript as Fig. 5a/b with accompanying discussion on their implications. The data were collected in collaboration with Marijo Parcina, MD of the Department of Micorbiology of the University Hospital Bonn in Bonn.

Reviewer #2

1. The Discussion needs to be extended to provide clearer interpretations of the findings in the study

Response:

We thank the reviewer for the suggestions and extended the discussion as highlighted in the revised manuscript. We extended the discussion about the effects of endotoxins as well as the suggestions made in point 3. and 5.

2. The conclusion on line 288 that wire injury mediated endothelial damage is not supported by robust quantitative data. The SEM in Fig 2 shows single images of endothelial damage or fibrin deposition and reports that similar areas were seen in 6 other mice. The authors should clarify how an unbiased and reproducible assessment of endothelial damage or fibrin deposition was carried out in n=6 mice. Do other findings in the study support the conclusion that endothelial damage has occurred

The motivation to use SEM for detecting endothelial damage rather than HnE staining or immunohistochemistry should be included in the text. The strengths and weaknesses of this strategy should be described in the Discussion. Do other studies show similar SEM images of endothelial damage?

Response:

We thank the reviewer for raising these important points, which provide an opportunity to clarify and strengthen our study.

To ensure an unbiased and reproducible assessment of endothelial damage and fibrin deposition, the examiners were blinded to the intervention and analyzed multiple regions per sample (6 regions per valve). We included n=6 mice to confirm consistency. Endothelial damage was evaluated based on the abundance of fibrin and destruction of the valvular endothelial cell layers. Representative SEM images were selected to illustrate typical findings, as reflected in Figure 2. We have now added a more detailed description of our sampling methodology to the Methods section (lines 176–179) to improve clarity.

SEM was chosen for its ability to provide high-resolution visualization of the endothelial surface, offering a three-dimensional perspective on structural changes, which complements traditional staining methods (line 388-392). We acknowledge that H&E staining and immunohistochemistry (IHC) are widely used techniques, and their application could provide additional insights into cellular and molecular-level changes. However, SEM enabled us to specifically focus on surface-level features, such as endothelial denudation and fibrin deposition, with clarity that H&E staining or IHC might not offer. Furthermore SEM allowed us to assess the valve damage including fibrin deposition and facilitates for visualization of bacterial infiltration and their localization on the aortic leaflets. It is a rare but targeted used technique to visualize valvular pathophysiologies as it allows the imaging of cells and the ECM with up to ×30,000 magnification, providing the high-quality visualisation of nuclei, the cell membrane, mitochondria, endoplasmic reticulum, Golgi apparatus, and engulfed fragments of the ECM .

Kostyunin et al. show valvular damage performing SEM analyses (compare Fig 3 of Kostyunin et al. Int. J. Mol. Sci. 2023; PMID: 37686408).

3. On lines 105 the authors state that "induction rate was 22% (104 CFU), 86% (105 CFU) and 92% (106 CFU)". The significant hop from 22% to 86% followed by a relative plateau should be discussed and interpreted in the Discussion

Response:

We thank the reviewer for the suggestions and extended the discussion relating to this topic (line 360-365). “104 CFU of S. aureus were unable to induce IE in both groups and IE induction plateaued between 105 and 106 CFU of S. aureus. Since all mice in the SEM showed evidence of endothelial damage and were thus susceptible to IE development, it can be assumed that immunomodulatory effects influenced the animals that did not develop endocarditis. These effects should be further explored in the future, including the impact of the microbiome, autoreactive effects, and early immune response activation (22-24).”

4. It is not clear why the immunohistochemistry of S .aureus in Fig 1i is performed up to 7 days which is in contrast to all other time frames in the same Figure. This should be clarified and motivated in text. How can the data in Fig 1g, h and i be interpreted in relation to one another?

Response:

Immunofluorescence imaging of S. aureus was performed 1, 3 and 7 days after bacterial challenge while blood and valvular cultures were obtained from samples 24h after bacterial challenge. We used blood cultures to test for bacteremia as a prerequisite for IE. Valvular cultures and S. aureus immunofluorescence staining of the valves confirmed IE induction in addition to the echocardiography assessments. The reliable induction of IE is the central finding in our manuscript and we decided to confirm bacterial infiltration of heart valves using two different methods: Vavlular cultures and S. aureus immunofluorescence staining aimed to use this approach to investigate whether the extent and location of bacterial infiltration change over time, or if complications such as abscess formation or aortitis develop. The latter could not be demonstrated, and the location of the bacteria did not change over time.

5. the wealth of data shown for echocardiography in Fig 3 should be better explained for those not familiar with the technique. Is all data important to include here or can critical parameters be highlighted and focused on in interpretation of the overall findings?

Response:

We could detect indirect signs of IE and endothelial damage using echocardiography. These include increase aortic cusp diameter, increased ventricular volumes and aortic regurgitation for IE and increased aortic peak velocity for AS. We specified the echocardiographic findings more closely in the discussion adding to its length and explained the technique and its findings more in detail in the Methods section.

---

## [Decision Letter · Decision Letter 1]

24 Jan 2025

A novel approach to studying Infective Endocarditis: Ultrasound-guided wire injury and bacterial challenge in mice

PONE-D-24-31793R1

Dear Dr. Bartsch,

We’re pleased to inform you that your manuscript has been judged scientifically suitable for publication and will be formally accepted for publication once it meets all outstanding technical requirements.

Kind regards,

Utpal Sen, Ph.D.

Academic Editor

PLOS ONE

Additional Editor Comments (optional):

Reviewers' comments:

Reviewer's Responses to Questions

**Comments to the Author**

1. If the authors have adequately addressed your comments raised in a previous round of review and you feel that this manuscript is now acceptable for publication, you may indicate that here to bypass the “Comments to the Author” section, enter your conflict of interest statement in the “Confidential to Editor” section, and submit your "Accept" recommendation.

Reviewer #1: All comments have been addressed

Reviewer #2: All comments have been addressed

2. Is the manuscript technically sound, and do the data support the conclusions?

Reviewer #1: Yes

Reviewer #2: Yes

3. Has the statistical analysis been performed appropriately and rigorously? 

Reviewer #1: Yes

Reviewer #2: Yes

4. Have the authors made all data underlying the findings in their manuscript fully available?

Reviewer #1: Yes

Reviewer #2: Yes

5. Is the manuscript presented in an intelligible fashion and written in standard English?

Reviewer #1: Yes

Reviewer #2: Yes

6. Review Comments to the Author

Reviewer #1: (No Response)

Reviewer #2: All of my comments have been adequately addressed in the resubmitted version of the manuscript and response to referee

7. PLOS authors have the option to publish the peer review history of their article (what does this mean? ). If published, this will include your full peer review and any attached files.

**Do you want your identity to be public for this peer review?** For information about this choice, including consent withdrawal, please see our Privacy Policy .

Reviewer #1: **Yes: ** Francesco Nappi

Reviewer #2: No

---

## [Editor Report · Acceptance letter]

PONE-D-24-31793R1

PLOS ONE

Dear Dr. Bartsch,

I'm pleased to inform you that your manuscript has been deemed suitable for publication in PLOS ONE. Congratulations! Your manuscript is now being handed over to our production team.

Kind regards,

on behalf of

Dr. Utpal Sen

Academic Editor

PLOS ONE